# Toward a better definition of hematopoietic progenitors suitable for B cell differentiation

Florian Dubois[1], Anne Gaignerie[2], Léa Flippe[1], Jean-Marie Heslan[1,2], Laurent Tesson[1], Mélanie Chesneau[1], Fabienne Haspot[1], Sophie Conchon[1], Laurent David[1,2], Sophie Brouard[1]*

**1** Université de Nantes, CHU Nantes, Inserm, Centre de Recherche en Transplantation et Immunologie, UMR 1064, ITUN, Nantes, France, **2** Université de Nantes, CHU Nantes, Inserm, CNRS, SFR Santé, FED 4203, Inserm UMS 016, CNRS UMS 3556, Nantes, France

* sophie.brouard@univ-nantes.fr

**Data Availability Statement:** All relevant data are within the manuscript and its Supporting Information files.

## Abstract

The success of inducing human pluripotent stem cells (hIPSC) offers new opportunities for cell-based therapy. Since B cells exert roles as effector and as regulator of immune responses in different clinical settings, we were interested in generating B cells from hIPSC. We differentiated human embryonic stem cells (hESC) and hIPSC into B cells onto OP9 and MS-5 stromal cells successively. We overcame issues in generating CD34+CD43+ hemato-poietic progenitors with appropriate cytokine conditions and emphasized the difficulties to generate proper hematopoietic progenitors. We highlight CD31intCD45int phenotype as a possible marker of hematopoietic progenitors suitable for B cell differentiation. Defining pre-cisely proper lymphoid progenitors will improve the study of their lineage commitment and the signals needed during the *in vitro* process.

## Introduction

B cells are important actors of immunity notably via their humoral responses with antibody production but also as antigen presenting cells leading to T cell activation [1]. B cells also display regulatory functions via cytokine production, cell-to-cell contact and by promoting regulatory T cells [2]. In humans, B cells with suppressive functions have thus been demonstrated in different clinical settings, including cancer, autoimmunity and tolerance towards kidney allografts [3, 4].

We showed that such B cells with regulatory properties were increased in patients with long term graft acceptance [5]. These regulatory B cells (Breg(s)) were able to efficiently block effector T cell proliferation via production of granzyme B and in a contact-dependent manner [5]. Other Bregs have been shown to display suppressive functions on different cell types and through multiple mechanisms, rendering Bregs attractive for therapy in different conditions [6]. However, Breg based-immunotherapy and their potential use in the clinic suffer several limitations. First, their use would need to rely on their clear characterization and the possibility of isolating them and to date, no consensual phenotype has been proposed thus rending their clinical application very difficult. Second, Bregs are small populations in humans and their expansion in vitro is very limited, and finally, their allogenic origin remains a major problem for clinical application.

**Funding:** This work was supported in the context of the Centaure project (RP10083) and the LabEX IGO thanks to French government financial support managed by the National Research Agency via the "Investment into the Future" program (ANR-11-LABX-0016-01). The ANR project BIKET (ANR-17-CE17-0008) also financially support this work. The funders had no role in study design, data collection and analysis, decision to publish, or preparation of the manuscript.

**Competing interests:** No authors have competing interests.

**Abbreviations:** hPSC, Human pluripotent stem cells; hIPSC, Human induced pluripotent stem cells; hES, Human embryonic stem cells; Bregs, Regulatory B cells; HSC, Hematopoietic stem cells; UCB, Umbilical cord blood; FCS, Foetal calf serum.

To circumvent these issues, we envisioned the use of hPSC, which are attractive in cell therapy given their capacity of self-renewal, to differentiate into multiple cell lineages and the possibility to choose the donor cells. B cell differentiation occurs in the bone marrow from HSC through different steps and a complex transcriptional process yet well characterized involving different cytokines [7, 8]. Previous rapport highlighted the efficiency of in vitro B cell differentiation from cord blood CD34$^+$ progenitors, as a source of HSC, on a monolayer of MS-5 or S17 stromal cells [9–11]. However, committing hPSC toward B cells remains a challenging process mainly due to the difficulty to generate proper hematopoietic stem cells from hPSC. Recent efforts have been made to generate proper hematopoietic stem cells from hPSC using different strategies, either by coculturing hPSC with murine bone marrow stromal cells or by the differentiation as 3D clusters of embryoid bodies [12, 13]. In this study, we tried to generate B cells from hPSC by implementing the available protocol [14] and we characterized these cells in comparison with B cells derived from UCB CD34$^+$ cells. We show that concomitant CD34 and CD43 expression by hematopoietic progenitors is not a phenotype describing a differentiation state sufficient before engaging B cell differentiation. We rather propose advanced progenitors displaying a CD31$^{int}$CD45$^{int}$ phenotypes as cells with proper lymphoid potential.

## Material and methods

### Human pluripotent stem cells, OP9 cells, MS-5 cells and UCB CD34$^+$ cells

hiPSC L71.019 [15], H9 hESC and H1 hESC [16] were cultured on Matrigel coated plates, in mTeSR1 media (Catalog 85850, STEMCELL technologies, Vancouver, Canada). Cells were weekly passed as clumps by a non-enzymatic method (Catalog 130-104-688, XF passaging solution, Miltenyi GmbH, Bergisch-Gladbach, Germany). Cells were regularly tested for mycoplasma. hESC were used under agreement from Agence de la Biomédecine RE17-007. hiPSC L71.19 were derived from fibroblasts obtained from Lonza (Catalog CC-2511, Lonza, Basel, Switzerland), originally prepared through anonymised skin samples.

OP9-GFP stromal cells were kindly provided by Juan-Carlos Zuniga-Pflucker, University of Toronto. OP9 cells were cultured in OP9 media consisted in αMEM (Catalog 22561–021, Gibco; Thermo Fisher Scientific, Inc., Waltham, MA, USA) containing 20% foetal calf serum (FCS, Catalog 10270–106, Gibco; Thermo Fisher Scientific, Inc.), 1% penicillin/streptomycin (Catalog 15140–122, Gibco; Thermo Fisher Scientific, Inc.) on gelatin coated plates.

MS-5 stromal cells were obtained from DSMZ (ACC 441, DSMZ, Brunswick, Germany). MS-5 cells were cultured in αMEM containing 10% FCS, 1% penicillin/streptomycin on gelatin coated plates. All cells were cultured at 37˚C under 5% CO2 and atmospheric O2 conditions.

Human umbilical cord blood (UCB) was obtained from Nantes University Hospital, after an informed and written consent. The human biological samples were collected during the patients care. These samples were integrated into the collections of human biological samples attached to the "Immunology" research program declared on 21/12/2012 under the n˚ DC-2012-1555 and in the following amending declarations (DC-2013-1832; DC2014-2206 and DC-2017-2987 currently pending) at the Ministry of Research and having obtained a favorable decision from the CPP Ouest IV on 06/03/2013. UCB CD34$^+$ cells were isolated by positive selection on magnetic columns according to the manufacturer's instructions (Catalog 130-046-702, CD34 MicroBead Kit, Miltenyi Biotec).

### B cell differentiation

The first step was to generate CD34$^+$ progenitors. Confluent hPSCs were harvested after dispase treatment (Catalog 07913, STEMCELL technologies) 5min at 37˚C. Then, 1–1.5x10$^6$ cells

were seeded as clumps in 10cm dishes on a monolayer of over-confluent OP9 stroma. Cells were cultured as previously described [14, 17] in αMEM containing 20% FCS, 1% penicillin/streptomycin and 100μM monothioglycerol (MTG) (Catalog M1753, Sigma-Aldrich, Saint-Louis, MO, USA). Total culture media was changed the first day to remove unattached cells and 50% of the media was systematically renewed at Day 4, 6 and 8. According to the different experiments, media was supplemented with BMP4 (Catalog 120-05ET, Peprotech, Rocky Hill, CT, USA), VEGF (Catalog 130-094-030), SCF (Catalog 130-096-695), Flt3L (Catalog 130-096-479) and IL-3 (Catalog 130-095-070) (all four from Miltenyi Biotec) respectively. At Day 10, cells were harvested by treatment with Trypsine-EDTA 0.05% during 15-20min at 37°C. After filtration through 40μm pores, CD34+ cells were isolated by positive selection on magnetic columns according to the manufacturer's instructions.

CD34+ cells were then seeded onto a monolayer of MS-5 stromal cells. MS-5 cells were previously irradiated at 50Gy one day before and seeded at a density of $5x10^5$ cells on gelatinised MW6 plates well. $5x10^4$ CD34+ cells were seeded per well in 4ml of culture medium and cultured for 21 days at 37°C. Culture media consisted in αMEM containing 10% FCS, 1% penicillin/streptomycin, 100μM MTG and SCF (50ng/ml), Flt3L (50ng/ml), IL-7 (20ng/ml, Catalog 130-095-363) and IL-3 (10ng/ml) (all from Miltenyi Biotec). Culture media was changed every 5 days with the previous cytokine except IL-3.

## Flow cytometry analysis

For CD34+ cell characterization, differentiated hPSCs at Day 10 and UCB CD34+ cells were stained with Fixable Yellow Dead Cell (Catalog L34967, Invitrogen; Thermo Fisher Scientific, Inc.) and the following antibodies CD34-PECy7 (Catalog 560710, clone 581; BD Biosciences, San Jose, CA, USA), CD43-APC (Catalog 560198, clone 1G10; BD Biosciences), CD31-PE (Catalog 555446, clone WM59; BD Biosciences) and CD45-APC-Cy7 (Catalog 557833, clone 2D1; BD Biosciences). For B cell characterization, cells were stained with Fixable Yellow Dead Cell (Invitrogen; Thermo Fisher Scientific, Inc.) and the following antibodies CD19-BUV395 (Catalog 563549, clone SJ25C1; BD Biosciences), CD10-BB700 (Catalog 745745, clone W8E7; BD Biosciences), CD45-PECy7 (Catalog 557748, clone HI30; BD Biosciences), CD34-BV711 (Catalog 745543, clone 8G12; BD Biosciences), IgM-FITC (Catalog 555782, clone G20-127; BD Biosciences) and CD79a-BV421 (Catalog 562852, clone HM47; BD Biosciences). Extracellular staining was performed during the viability staining in order to limit the loss of cells with $5.10^4$ cells in PBS during 20min at 4°C. Intracellular staining of CD79a was performed in Perm Wash Buffer (Catalog 88-8824-00, eBioscience, Vienna, Austria) (30 min, 4°C) after fixation/permeabilization (eBioscience) (30 min, 4°C). Supervised analysis was performed with FlowJo v.10 software on CD43, CD45 and CD31 on CD34+ gated cells after doublet and dead cells exclusion. Fluorescence minus one (FMO) was used to gate on marked cells.

## Gene expression analysis

At the different steps of the differentiation protocols, RNA was extracted from total cells with a minimum of $3.10^5$ cells using the RNeasy Kit (Catalog 74106, Quiagen, Hilden, Germany) according to the manufacturer's instructions. Reverse transcription was performed with 0.5μg of RNA according to the manufacturer's instructions (Catalog 4368814, High Capacity cDNA Reverse Transcription Kit, Applied Biosystems; Thermo Fisher Scientific, Inc). Expression analysis of *PAX5*, *VPREB1*, *IGLL1*, *IL-7RA*, *EBF1*, *RAG1*, *DNTT* and *POU5F1* (*OCT4*) was performed using TaqMan™ Gene Expression Assays (Applied Biosystems, Thermo Fisher Scientific, Inc). References of TaqMan probes are described in Table 1. PCR reactions were carried out with cDNA diluted to 1/5 in a final volume of 10μl to avoid PCR inhibition. Gene expression levels were

**Table 1.**

| Target genes | References |
| --- | --- |
| PAX5 | Hs00277134_m1 |
| VPREB1 | Hs00356766_g1 |
| IGLL1 | Hs00252263_m1 |
| IL7R | Hs00902334_m1 |
| EBF1 | Hs01092694_m1 |
| RAG1 | Hs00172121_m1 |
| DNTT | Hs00172743_m1 |
| POU5F1 (OCT4) | Hs00999632_g1 |

calculated using the $2^{-\Delta\Delta CT}$ method [18]. Gene expressions were reported to the expression of both the housekeeping genes *GAPDH* and *HPRT* and to a calibrator with undetermined CT or CT with a value higher than 35, set to 40. Gene expression analysis were also performed on tonsil B cells as control of B cell gene expression. Human tonsils were obtained from the Plateau technique médico-chirurgical (PTMC) of the Nantes University Hospital after an informed and written consent was obtained. Tonsil cells were mechanically isolated freshly after their reception. Total B cells were then purified by negative selection on magnetic columns according to the manufacturer's instructions (B Cell Isolation Kit II; Miltenyi Biotec).

## Statistical analyses

Statistical analyses were performed using GraphPad Prism (GraphPad Software, San Diego, CA). Significance of difference between nonparametric variables were assessed with a Mann-Whitney test and the differences were considered statistically significant at $p < 0.05$. All results are expressed as means ± SEMs.

## Results

### Assessing B cell differentiation

Our goal was to generate B cells from hPSCs. To do so, we started by implementing a published protocol, outlined in Fig 1A [14]. Briefly, hPSCs (H9 hESC and hIPSC L71.019) were differentiated toward CD34$^+$ hematopoietic progenitors during 10 days onto a monolayer of OP9 stromal cells. Then, CD34$^+$ cells were cultured onto a monolayer of MS-5 stromal cells for B cell differentiation for 21 days, in the presence of IL-7, IL-3, SCF and Flt3L. B cell differentiation was measured by the frequency of CD10$^+$CD19$^+$ B cells obtained after differentiation. CD34$^+$ cells isolated from umbilical cord blood (UCB) were cultured in parallel in the same conditions, as positive control. Starting from H9 hESC (Fig 1B) and from H1 hESC (S1A Fig), we obtained 5% of CD34$^+$ and they did not express CD43, a crucial intermediate phenotype for B cell differentiation [14]. This resulted in an impairment of B cell differentiation considering that less than 0.2% of CD10$^+$CD19$^+$ B cells were generated after 21 days of differentiation from these CD34$^+$ (Figs 1C and S1B). In contrast and using the same protocol and conditions, UCB CD34$^+$ cells efficiently engaged into CD10$^+$CD19$^+$ B cells (7.1%) at Day 21, without IgM expression (Fig 1C).

### Cytokines trigger the generation of CD34$^+$CD43$^+$ hematopoietic progenitors with low ability to differentiate into CD19$^+$ B cells

Considering these first data, we set out to improve the generation of CD34$^+$CD43$^+$ on OP9 monolayers during CD34$^+$ differentiation. Our strategy consisted in adding cytokines to the

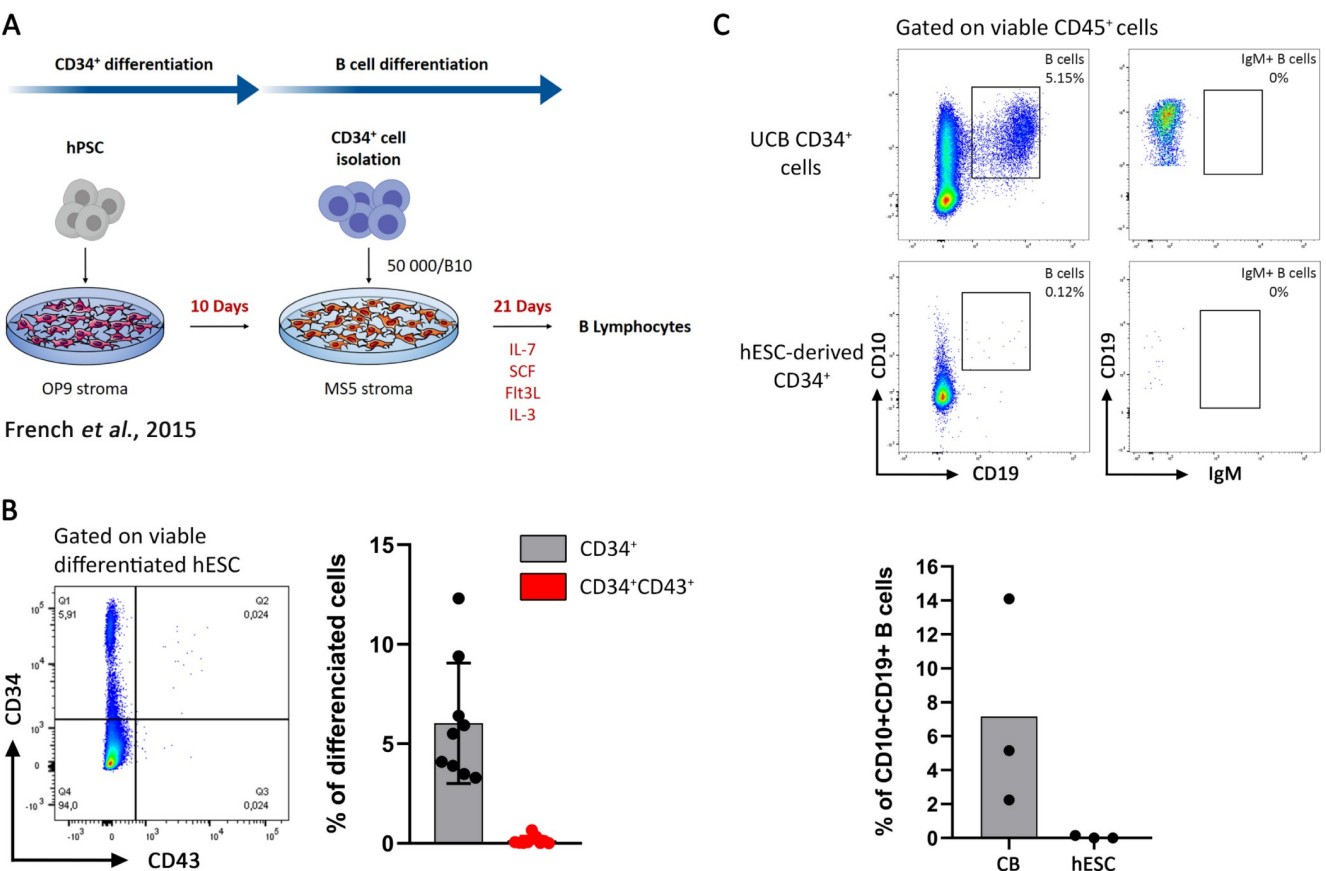

**Fig 1. CD34+ generation from hPSC without cytokines and their low potential in B cell differentiation.** (A) Protocol of hPSC differentiation into B cells as previously described [14]. (B) CD34+ differentiation from H9 hESC onto OP9 stroma at Day 10. CD34/CD43 expression analyzed by flow cytometry on viable differentiated hESC is shown in the left as a dot plot from one representative experiment and the proportion of CD34+ cells (in black) and CD34+CD43+ (in red) are shown on the right panel, n = 9 experiments. (C) B cells differentiation at Day 21 from UCB CD34+ cells (Upper panel) and CD34+-derived hESC (middle panel). CD10+CD19+ B cells proportion and their IgM expression was analyzed on viable CD45+ cells, dot plot from one representative experiment is shown. Histogram of CD10+CD19+ B cells proportion is presented in the lower panel (n = 3 experiments from both UCB CD34+ cells and hESC-derived CD34+).

OP9 media, to trigger optimal hematopoietic differentiation from hESC. VEGF, SCF and Flt3L were kept in the medium from Day 1 to Day 10 at 10ng/ml with or without BMP4 (5ng/ml) and IL-3 (10ng/ml) as detailed in Fig 2A. We found that a significant increase in CD34+CD43+ proportion using VEGF, SCF, Flt3L and BMP-4 combined with IL-3 from Day 4 to Day 10 (Fig 2B, p = 0.0364) from 0.05% to 0.8%. Interestingly, the use of VEGF, SCF and Flt3L combined with IL-3 from Day 4 to Day 10 and without BMP4 permits a higher improvement of CD34+ differentiation, increasing the yield of CD34+CD43+ up to 6%, not significantly at 10ng/ml because of the small sample size (Fig 2B) but significantly at 25ng/ml (S1C Fig, p = 0.0091). However, B-cell differentiation conducted with these progenitors until Day 21 resulted in 0.2% of CD10+CD19+ B cells (Fig 2C). These results were also obtained when all cytokines were set to 25ng/ml during the CD34+ differentiation (S1C and S1D Fig) suggesting that hESC-derived CD34+CD43+ cells were not in a state permissive to B cell differentiation.

To further improve our protocol of differentiation, we kept the same cytokine cocktail than the one allowing the best CD34+CD43+ differentiation (i.e. VEGF, SCF, Flt3L constant and combined with IL-3, from Day 4 to 10), and we tested different concentrations of the cytokines. Starting from H9 hESC, we found that the concentration of 25ng/ml significantly

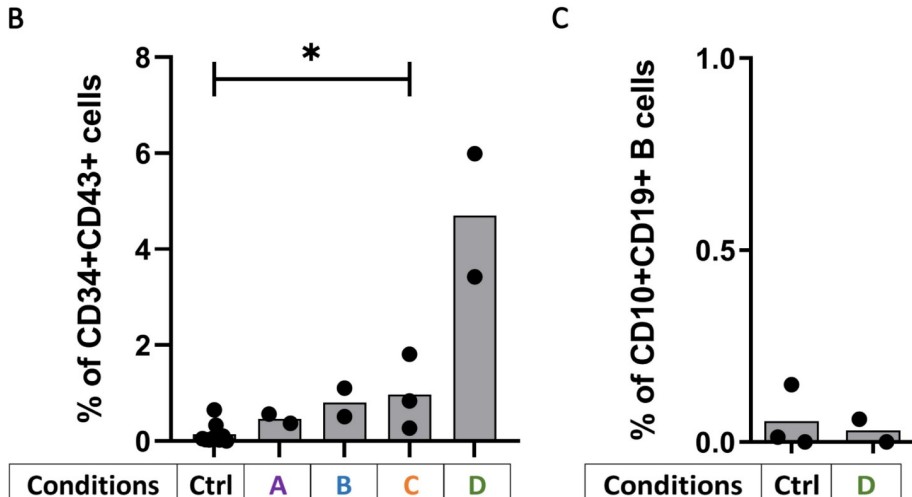

**Fig 2. Cytokines allow CD34$^+$CD43$^+$ differentiation from stem cells.** (A) Table of the cytokine conditions tested according their presence from Day 1 to Day 3 and Day 4 to Day 10 for the CD34$^+$ differentiation (cytokines were used at 10ng/ml except for BMP4 at 5ng/ml). (B) Percentage of CD34$^+$CD43$^+$ cells at the end of the CD34$^+$ differentiation according to the cytokine conditions used (n = 2 or 3 independent experiments) or without cytokines as control (Ctrl, n = 9 independent experiments). The significance of differences between the condition C and the Ctrl was determined using a Mann-Whitney Test; *p < 0.05. (C) Percentage of CD10$^+$CD19$^+$ B cells at Day 21 of B cell differentiation from hESC-derived CD34$^+$ with (Condition D; n = 2 independent experiments) or without cytokine (Ctrl, n = 3 independent experiments).

increase the proportion of CD34$^+$ (p = 0.0364) and CD34$^+$CD43$^+$ progenitors (p = 0.0091) (Fig 3A). However, while not significant because of the small sample size (n = 2), the concentration of 10ng/ml for our cytokine cocktail triggered the highest proportion of CD34$^+$ (22.8%) and CD34$^+$CD43$^+$ progenitors (4.7%) (Fig 3A), without increasing the final CD10$^+$CD19$^+$ B cell proportion (< 0.2%) (Fig 3B). Starting from hIPSC, there was no difference in CD34$^+$CD43$^+$ proportion according to the different concentrations of the cytokine cocktail used (Fig 3C), and as previously this resulted in only a low level of final CD10$^+$CD19$^+$ B cell proportion (0.7% at 10ng/ml, Fig 3D).

Altogether our results indicate that modifying cytokine concentrations efficiently increased the CD34$^+$CD43$^+$ proportion in H9 hESC but did not significantly improve the subsequent CD10$^+$CD19$^+$ B-cell differentiation suggesting that CD43 expression by CD34$^+$ progenitors is not sufficient to engage them toward B cell differentiation.

## hPSC-derived CD34$^+$ generated Pro-B and Pre-B cells on MS-5 stroma

During B cell differentiation, CD45$^+$CD10$^+$ common lymphoid progenitors (CLP) differentiate into CD79a$^+$CD19$^-$ Pre-Pro B cells and then into Pro-B and Pre-B cells included in the CD79a$^+$CD19$^+$ fraction. While using our protocol to generate CD34$^+$ with the same cytokine condition at 10ng/ml, we wanted to better follow and appreciate their B cell differentiation.

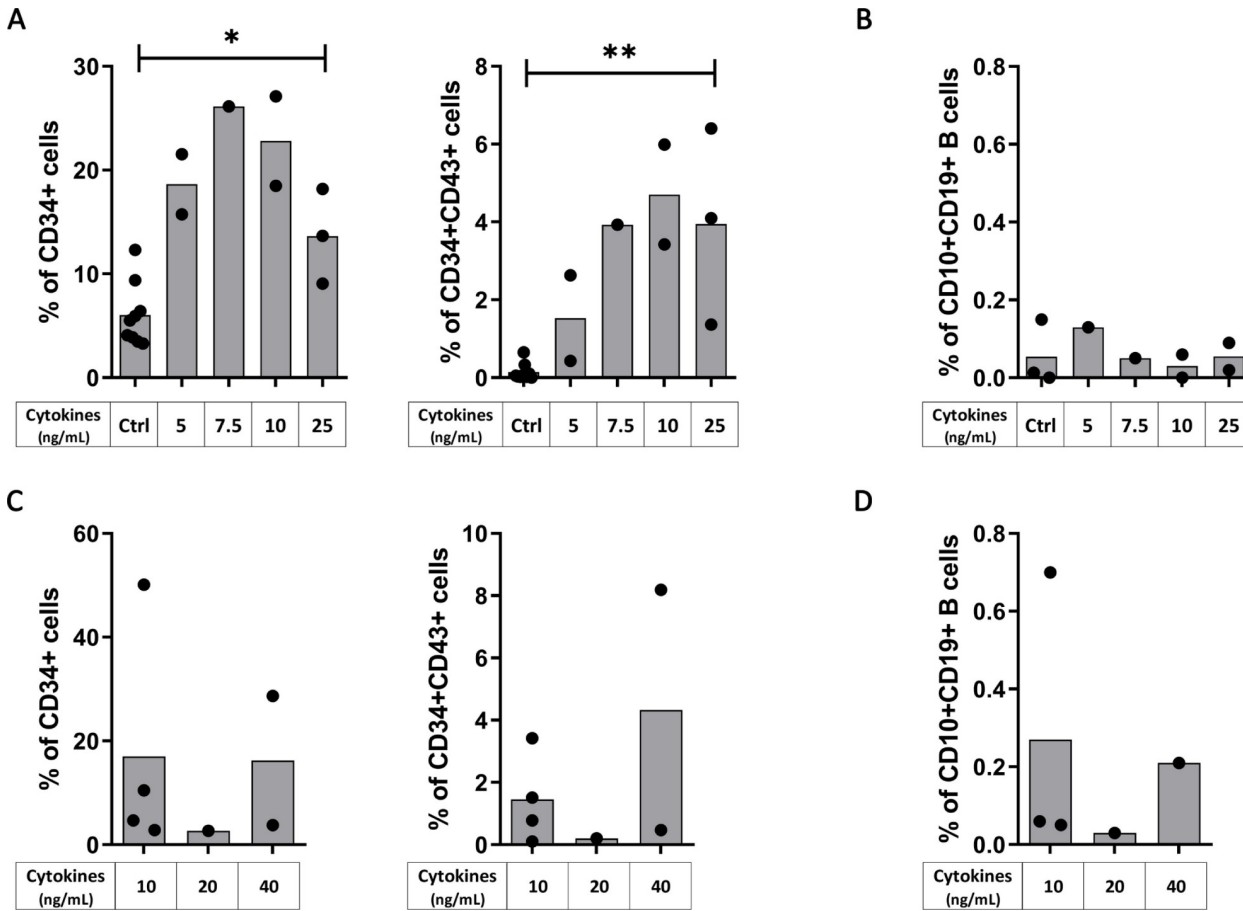

**Fig 3. Cytokines at 10ng/ml permit the best B cell differentiation.** hESC (Upper panels) and hIPSC (lower panels) were used for CD34$^+$ differentiation with our protocol of cytokines addition (the cytokine condition D, as detailed in Fig 2A) at different concentration. (A, C) Histograms show the proportion of CD34$^+$ (left panel) and CD34$^+$CD43$^+$ (right panel) after CD34$^+$ differentiation from (A) hESC with cytokines (n = 1 to 3 independent experiments) or without cytokines as control (Ctrl, n = 9 independent experiments) and from (C) hIPSC (n = 1 to 4 independent experiments) The significance of differences between the condition at 25ng/mL and the Ctrl was determined using a Mann-Whitney Test; $^*$p < 0.05; $^{**}$p < 0.01. (B, D) B cells proportion at Day 21 after B cell differentiation from the corresponding CD34$^+$ cells is shown from (B) hESC-derived CD34$^+$ cells with cytokines (n = 1 or 2 independent experiments) or without cytokines as control (Ctrl, n = 3 independent experiments) during CD34$^+$ differentiation and from (D) hIPSC-derived CD34$^+$ (n = 1 to 3 independent experiments).

We thus analyzed during B cell differentiation, CD10$^+$CD19$^+$ B cell proportion among CD45$^+$ cells and we looked at CD79a expression among the CD45$^+$CD10$^+$ lymphoid progenitors allowing us to distinguish CLP which are CD79a$^-$CD19$^-$, CD79a$^+$CD19$^-$ Pre-Pro B cells and the CD79a$^+$CD19$^+$ fraction composed of the Pro and Pre-B cells as detailed in Fig 4A. Starting B cell differentiation from UCB CD34$^+$ cells, we obtained 30% CLP, 50.74% Pre-pro B cells and 15% Pro-B and Pre-B cells among the CD45$^+$CD10$^+$ cell population (Fig 4B). During B cell differentiation of hESC-derived CD34$^+$, we observed that all CD45$^+$CD10$^+$ were CLP at Day 7 and their frequency decreased to 82% at Day 21. Pre-pro B cells appeared between Day 7 and Day 14 and increased from 6.47% to 10.9% of CD45$^+$CD10$^+$ cells, from Day 14 to Day 21. Finally, the fraction consisting of Pro-B cells and Pre-B cells represented 1.25% of CD45$^+$CD10$^+$ at Day 14 and 10.7% of CD45$^+$CD10$^+$ at Day 21 (Fig 4C). These data show that CD34$^+$ hematopoietic progenitors were engaged into B lymphopoiesis reaching the stage of Pro-B and Pre-B cells, albeit in limited amounts (S3 Fig).

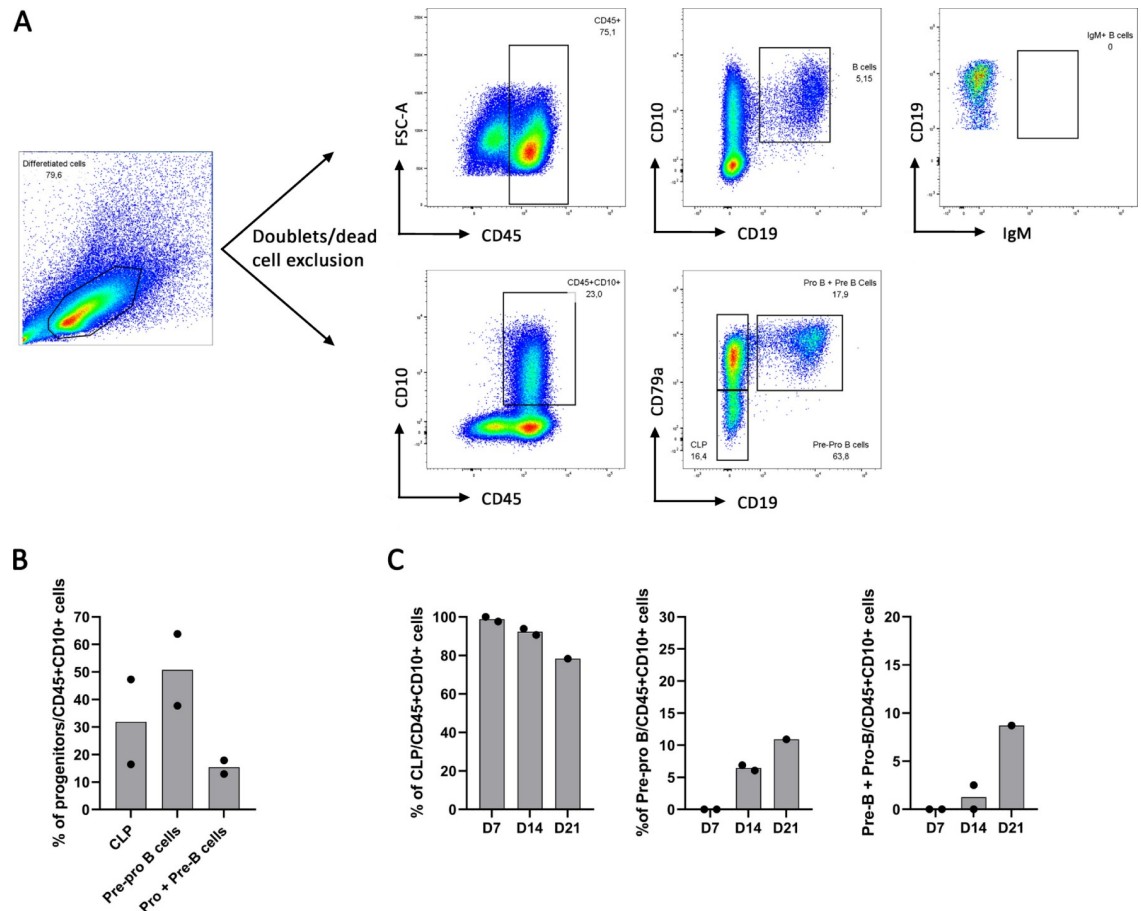

**Fig 4. Gating strategy of B cell progenitors.** (A) B cell characterization during B cell differentiation of UCB CD34$^+$ cells using CD10$^+$CD19$^+$ total B cells gated on CD45$^+$ (upper panel) and using CD79a$^-$CD10$^-$ CLP, CD79a$^+$CD10$^-$ Pre-pro-B cells and CD79a$^+$CD19$^+$ both Pro-B and Pre-B cells gated on CD45$^+$CD10$^+$ (lower panel). (B) B cell progenitors proportion in 2 independent experiments. (C) B cell progenitors proportion at Day 7 (D7), 14 (D14) and 21 (D21) of B cell differentiation from hESC-derived CD34$^+$ (n = 1 or 2 independent experiments).

### hESC-derived cells express genes implicated in B cell commitment during B cell differentiation

We analyzed gene expression known to be important for B cell commitment in the course of B cell differentiation from CD34$^+$ progenitors after 7, 14, and 21 days of culture on MS-5 stroma. We thus extracted RNA of total cells at these different time points and analyzed the transcripts expression of the following genes *PAX5*, *EBF1*, *VPREB1*, *IGLL1*, *IL-7RA* and *RAG1* (Fig 5). While *OCT4* (*POU5F1*) was used as pluripotency control, undifferentiated H9 hESC and tonsil B cells served as control of gene expression. At Day 21, B cell differentiation from UCB CD34$^+$ cells was characterized by a low gene expression for *PAX5*, *EBF1*, *IL-7RA* and *OCT4* and a high gene expression for *VPREB1*, *IGLL1* and *RAG1* (Fig 5A). Regarding differentiated hESC from Day 7 to Day 21, they harbor a low *PAX5* gene expression, a decreasing gene expression for *EBF1* and a slight increase for *IL-7RA* with their overall expression level similar to those obtained from UCB CD34$^+$ cells (Fig 5B). We observed an expression of *VPREB1* gene starting at Day 14 while *IGLL1* gene expression is present at Day 7 and decreased during the differentiation. This is associated with an increase in *RAG1* expression from Day 7 to Day 21, known to be important for immunoglobulin (Ig) gene rearrangement leading to the formation of the

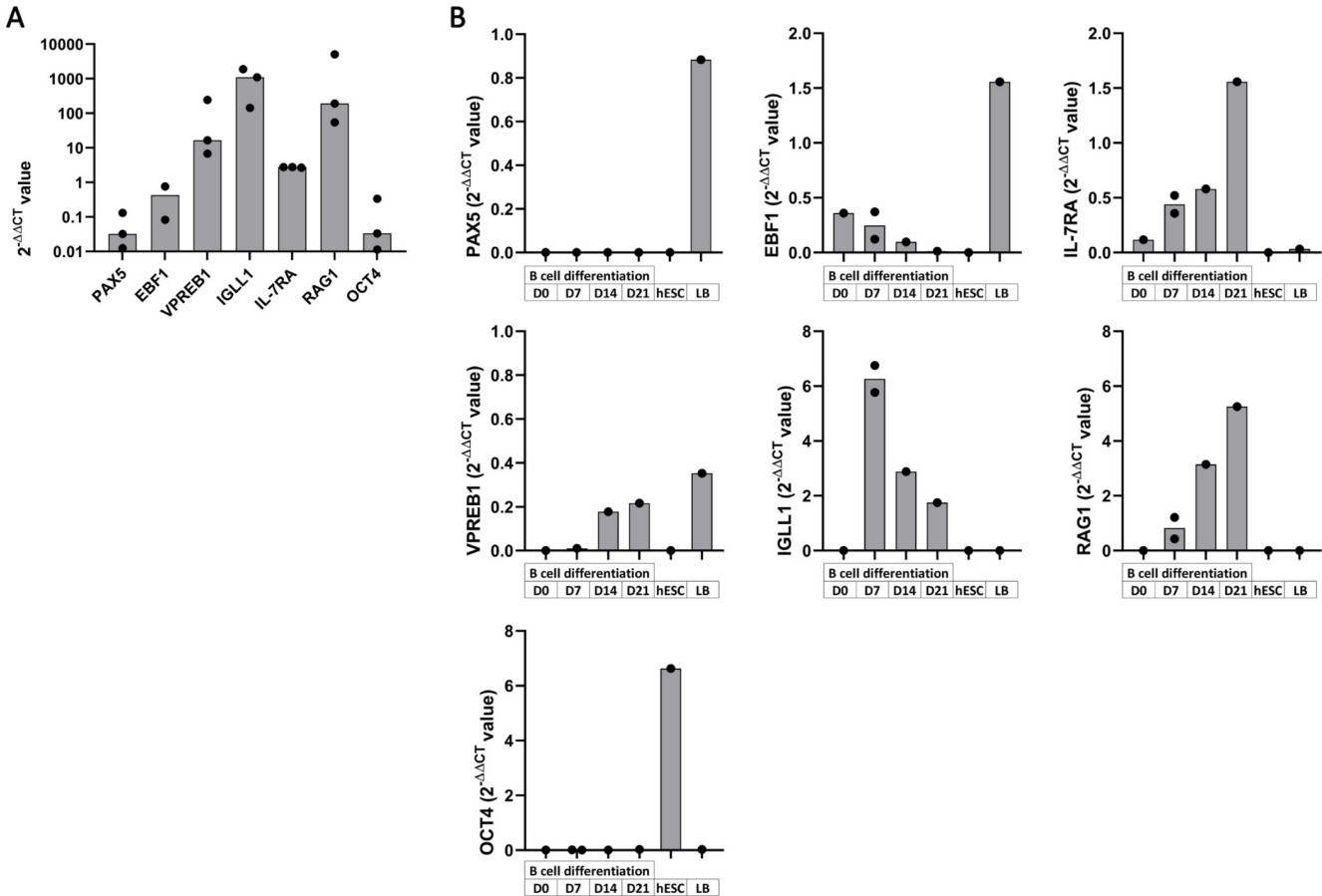

**Fig 5. Gene expression at different time point of B cell differentiation.** Gene expression analysis by Taqman expression assay of *PAX5*, *EBF1*, *VPREB1*, *IL-7RA*, *IGLL1*, *RAG1* and *OCT4* were performed on total cells at different time point during B cell differentiation. Histograms of $2^{-DDCT}$ value display gene expression analysis of cells (A) at Day 21 of B cell differentiation from UCB CD34$^+$ cells (n = 2 to 3 independent experiments) and (B) at Day 0 (D0), 7, 14 and 21 of B cell differentiation from hESC-derived CD34$^+$ with undifferentiated hESC and tonsil B cells (LB) as controls (n = 1 to 2 independent experiments).

Pre-BCR. *IL-7RA*, *IGLL1* and *VPREB1* expression levels were lower than during B cell differentiation from UCB CD34$^+$ cells, this could be explained by the lower number of B cells generated from hESC. Finally, during B cell differentiation from hESC-derived CD34$^+$ cells, *OCT4* was not expressed compared to the undifferentiated hESC consistent with their exit from pluripotency. Thus, while we obtained few B cells from H9 hESC, these results are concordant with a normal process of early B cell differentiation (S3 Fig).

## UCB CD34$^+$ cells exhibit a CD31$^{int}$CD45$^{int}$ phenotype and differentiate into CD19$^+$ B cells

The CD34$^+$ differentiation is a crucial step to obtain hematopoietic progenitors in sufficient amount and abilities to generate B cells on MS-5 stroma. While UCB CD34$^+$ cells succeeded to allow great B cell differentiation on MS-5 stroma (7%), CD34$^+$ generated from both hESC and hIPSC did not reach this efficiency. To better characterize the progenitors that could differentiate into B cells, we extended our profiling of the CD34$^+$ cells and we measured the expression of CD31, an endothelial marker and CD45, a marker of hematopoietic cells expressed after CD43 during hematopoiesis on UCB CD34$^+$ cells. We observed that UCB CD34$^+$ cells

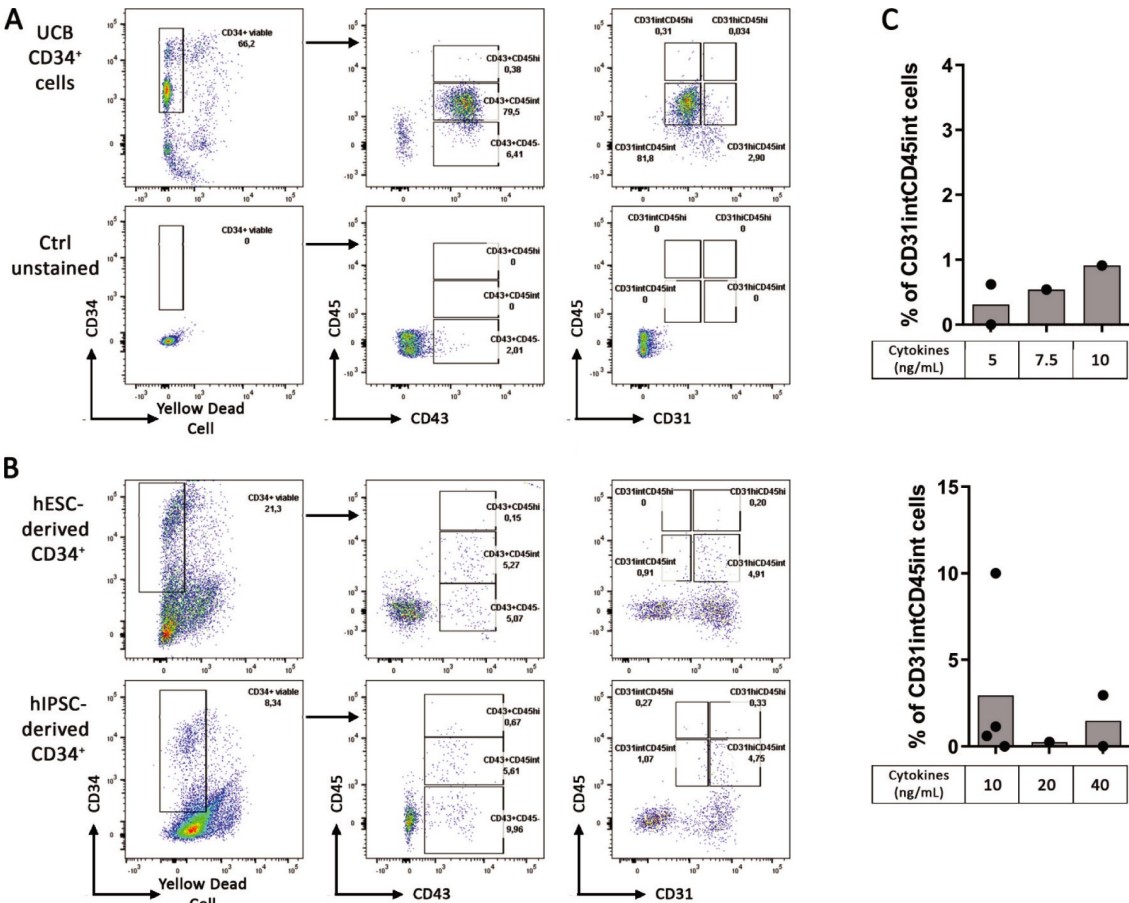

**Fig 6. CD31$^{int}$CD45$^{int}$ are important progenitors for B cells differentiation.** Characterization of CD34$^+$ progenitors by their expression analysis of the following markers CD45, CD43 and CD31. FACS analysis was performed from (A) UCB CD34$^+$ cells (Upper panel) with unstained cells as control (lower panel), dot plot representative of 2 experiments and from (B) hESC-derived CD34$^+$ (Upper panel) and hIPSC-derived CD34$^+$ (lower panel) using our CD34$^+$ differentiation protocol with cytokines at 10ng/ml, dot plot representative of 2 experiments for hESC and 4 experiments for hIPSC. (C) Histograms show CD31$^{int}$CD45$^{int}$ proportion in hESC (upper panel) and hIPSC (lower panel) at Day 10 of CD34$^+$ using our CD34$^+$ differentiation protocol with cytokines at different concentrations (n = 1 to 4 independent experiments).

expressed CD43 and were CD31$^{int}$CD45$^{int}$ (Fig 6A). In contrast, CD34$^+$ differentiated from hESC and hIPSC using our same protocol as before with cytokines at 10ng/ml, expressed few CD45 and were mainly CD31$^{hi}$ (Fig 6B). In addition, variation in cytokine concentrations did not increase CD31$^{int}$CD45$^{int}$ proportion in CD34$^+$ differentiated neither from hESC nor hIPSC (Fig 6C). These results on UCB CD34$^+$ cells suggest that hematopoietic progenitors possess a CD31$^{int}$CD45$^{int}$ phenotype that should be reached by differentiated hPSC before engaging B cell differentiation.

On this basis, we hypothesized that CD34$^+$ progenitors differentiated from hPSC were not at an optimal differentiation stage before engaging the B cell differentiation. Therefore, we stopped the CD34$^+$ differentiation from hIPSC at 8, 10 and 12 days and compared the CD34$^+$ progenitor's ability to differentiate into B cells. As expected, we observed an increase in the proportion of CD43$^+$CD45$^{int}$, CD43$^+$CD45$^{hi}$, CD31$^{int}$CD45$^{int}$ and a decrease in the proportion of CD43$^+$CD45$^-$ from Day 8 to Day 12 of CD34$^+$ differentiation (Fig 7A). Interestingly, the proportion of CD31$^{hi}$CD45$^{int}$ increased faster than the proportion of CD31$^{int}$CD45$^{int}$ and reached a similar proportion at Day 12. However, despite a higher proportion of CD31$^{int}$CD45$^{int}$ at Day

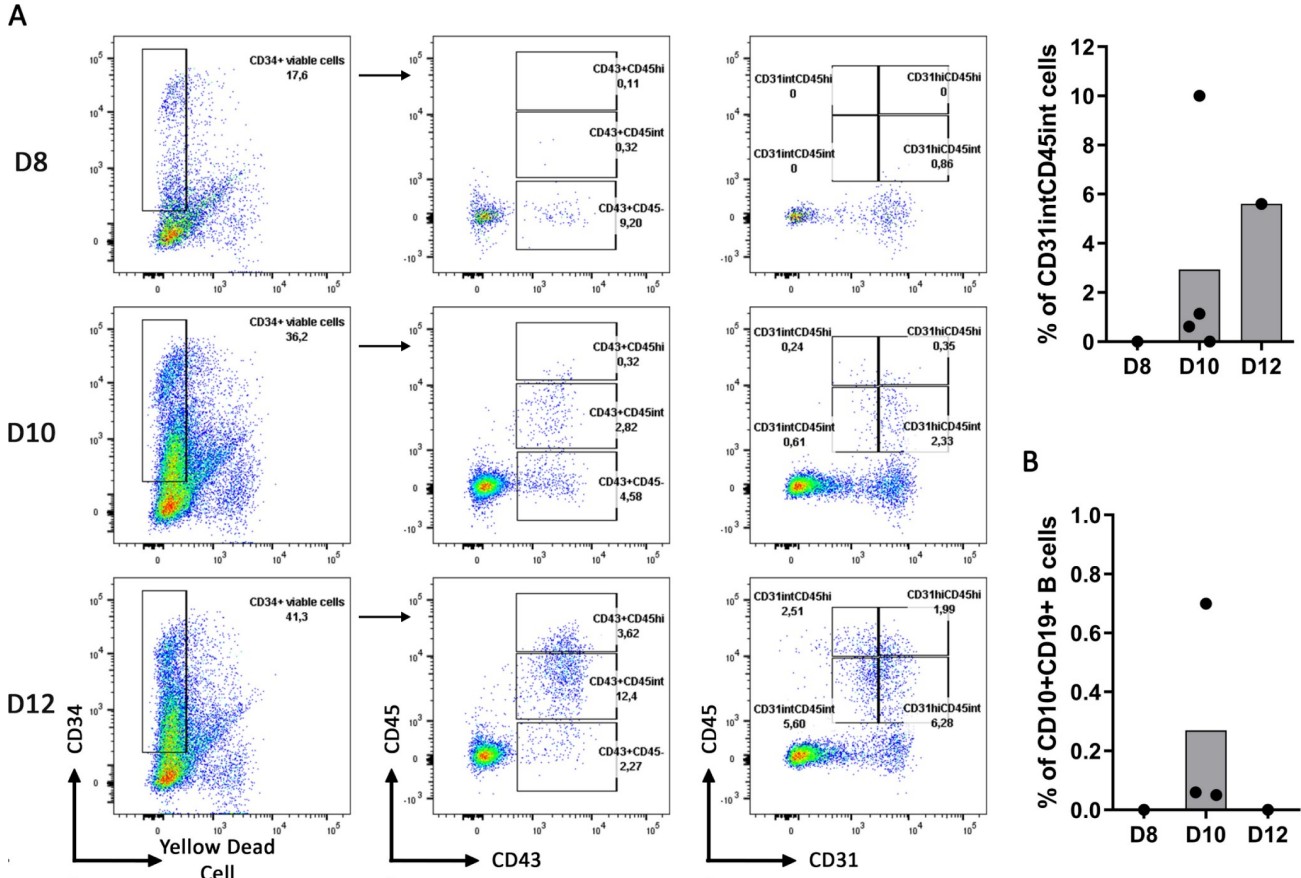

**Fig 7. Day 10 of CD34+ differentiation permits the better B cell differentiation.** (A) CD34+ differentiation from hIPSC, with our protocol of cytokine addition (cytokine condition D, as detailed in Fig 2A) at 10ng/ml, was stopped at Day 8, Day 10 and Day 12 to analyze the expression of CD34+ cells for the following markers CD45, CD43 and CD31. Dot plots from one experiment for Day 8 and Day 12 and 4 independent experiments for Day 10 are shown in the left panel. Histogram of the CD31intCD45int proportion among the CD34+ at different time of the CD34+ differentiation is shown. (B) Histogram represents the B cell proportion at the end of the B cell differentiation using the corresponding hIPSC-derived CD34+ (n = 1 to 3 independent experiments).

12, CD10+CD19+ B cells proportion was not increased upon B cell differentiation from these hIPSC-derived CD34+ cells (Fig 7B).

Altogether, our results point out the difficulties to obtain hematopoietic progenitors with strong lymphoid abilities from hPSC. We obtained only low B cell proportion from hESC and hIPSC. We show that while CD43 expression is needed in hPSC-derived CD34+ cells, CD43 is not sufficient for differentiation into B cells. UCB CD34+ cells easily differentiate into B cells and harbor a CD31intCD45int phenotype.

## Discussion

Human pluripotent stem cells display unlimited self-renewal capacities and give rise to all cell types of the body. It is well accepted that during hematopoiesis in the embryo, the first wave starting in the extra-embryonic yolk sac gives rise to transient hematopoietic progenitors with differentiation potential restricted for primitive erythrocytes, megakaryocytes, and macrophages [19, 20]. Then, the definitive wave leads to adult HSC production with mature lymphoid-myeloid differentiation potential in the aorta-gonad-mesonephros (AGM). These HSC transit to the fetal liver before residing in the bone marrow for hematopoiesis during adult life

[21]. It is also accepted that adult HSC are produced from endothelial cells with hematopoietic capacities called hemogenic endothelium (HE). This HE possesses endothelial phenotype but undergoes endothelial-to-hematopoietic transition (EHT) to generate HSC that further differentiate into hematopoietic progenitors. The *in vitro* hematopoietic differentiation from both hESC and hIPSC has been described in numerous studies with multiple blood lineage formation [13]. Considering the dual role of B cells as effector and regulator of immune responses [22, 23] and their involvement in many pathological settings [3, 5, 24], we aim to attempt B cell differentiation from hIPSC. Indeed, hIPSC generation from differentiated cell types represents a real breakthrough in stem cell research and allow important opportunities for disease modeling, drug screening and cell-based therapy [25]. Thus, B cell generation from hIPSC would offer several advantages including an unlimited cell source, the choice of cell donor origin and the possibility to modulate and control their gene expression in order to generate B cells with regulatory properties and potential clinical use.

French *et al.* succeeded in generating B cells from hIPSC by first culturing them onto OP9 stromal cells for 10 days without cytokines to generate CD34+ progenitors. CD34+ sorted cells were then cultured for 21 days onto MS-5 stromal cells with cytokines to reach 5% of CD10+CD19+ B cells [14]. In this study, we tried to differentiate B cells from hESC and hIPSC following this protocol. While we were able to generate CD34+ cells from both hESC and hIPSC, these progenitors were not able to generate more than 0.2% B cells onto MS-5 stroma. In our hands, generated CD34+ cells lacked CD43 expression, which is the first marker expressed during hematopoiesis [26] but also necessary for B cell differentiation [14]. Different strategies have been published in the literature to generate CD34+ hematopoietic progenitors from hPSC by their differentiation as 3D clusters of embryoid bodies in feeder-free conditions with cytokines [27–29]. To improve CD34+CD43+ generation, we tested the effect of cytokine addition during the 10 days coculture model of hIPSC onto OP9 stromal cells. Addition of cytokines allowed the generation of CD34+CD43+ and surprisingly the optimal condition was without BMP4. This is not consistent with the fact that BMP4 is important for the mesodermal differentiation prior EHT leading to hematopoietic progenitor generation [28, 30, 31]. This could be due to the fact that OP9 stromal cells alone support mesodermal generation of hPSC and the addition of BMP4 might not synergize with this OP9 stroma ability.

Contrasting with French *et al.* data, and despite our ability to generate CD34+CD43+ cells either from hESC or hIPSC, no more than 0.7% B cells were differentiated at the end of the 21 days onto MS-5 stroma and this was not dependent on the frequency of CD34+CD43+ cells present in the culture prior to B cell differentiation. It suggests that CD43 expression is not sufficient as a marker of CD34+ progenitors suitable for B cell commitment *in vitro*. We thus investigated differences in hPSC-derived CD34+ and UCB CD34+ cells considering the easier commitment into B cells of the latter ones. While all the UCB CD34+ cells expressed CD43 and intermediate levels of CD31 and CD45, the majority of CD34+ derived from hESC and hIPSC were CD31hi CD45int, suggesting that these cells were at different hematopoiesis stages. In addition, CD31intCD45int cells were absent at Day 8 of CD34+ differentiation from hIPSC, they appeared between Day 8 and Day 10 and their proportion increased from Day 10 to Day 12 suggesting hematopoietic progenitors were first CD31hi and lost CD31 expression during differentiation. However, even with this higher CD31intCD45int proportion at Day 12, it did not increase B cell differentiation efficiency. This might be explained by the fact that these CD31intCD45int were not enrich in the total CD45+ cell proportion which also increase at Day 12 of CD34+ differentiation. Indeed, in one experiment with 10% of the CD31intCD45int, which represent the majority of CD45+, we obtained our highest B cell generation (S2 Fig). Thus, it suggests that CD31intCD45int cells could be important progenitors for B cell

differentiation and that CD34$^+$ differentiation should be oriented toward an enrichment of CD31$^{int}$CD45$^{int}$ cells among CD45$^+$ cells for this engagement.

B cell generation needs specific environment and gene activation to trigger their differentiation from hematopoietic stem cells [32]. *PAX5* and *EBF1* are two transcription factors that specifically drive and maintain a B cell commitment during hematopoiesis [33]. The Pro-B cells, which first express CD19, also express the recombination-activated gene (*RAG1* and *RAG2*) for the V(D)J rearrangement of the immunoglobulin heavy chain (Igh) locus. This Igh associates to the surrogate light chain composed of the two subunits iota polypeptide chain (VPREB) and the Immunoglobulin lambda-like polypeptide 1 (IGLL1, λ5) to form the Pre-BCR. The pre-BCR is needed for the positive selection of Pre-B cells [34] resulting in the immunoglobulin light chain rearrangement and their transition to immature IgM$^+$ B cells that we attempt to generate. At Day 21 of B cell differentiation from UCB CD34$^+$ cells and hESC-derived CD34$^+$ cells, small amounts of *PAX5* and *EBF1* were observed but were sufficient for B cell commitment. These differentiated cells also expressed low level of *IL-7RA*, important for B cell differentiation. We also found during B cell differentiation from hESC, an expression of *IGLL1* and *VPREB1* associated with an increased *RAG1* expression from Day 7 to Day 21 of B cell differentiation. At Day 21, expressions of *IGLL1*, *VPREB1* and *RAG1* were 10 to 100-fold lower than in cells after B cell differentiation from UCB CD34$^+$ cells. While this highlights again the more efficient commitment of UCB CD34$^+$ cells into B cells and the higher number of B cells generated, it shows that gene expressions in cells after our protocol of B cell differentiation from hPSC are concordant with a B cell differentiation process.

In parallel, the phenotype of B cells and their differentiation rate was followed by flow cytometry. During B cell differentiation process in the bone marrow, CD10$^+$CD79a$^-$CD19$^-$ CLP first differentiate into CD10$^+$CD79a$^+$CD19$^-$ Pre-Pro B cells which successively differentiate into CD10$^+$CD79a$^+$CD19$^+$ Pro-B cells and Pre-B cells. Starting from UCB CD34$^+$ cells, we found that 50% of CD45$^+$CD10$^+$ cells were Pre-Pro B cells at Day 21, 30% were CLP and 15% were Pro and Pre-B cells. Starting from hESC-derived CD34$^+$ cells, we observed from Day 7 to 21 of B cell differentiation, a decrease of the CLP proportion (99 to 78.%) and an increase of Pre-Pro B cells (0 to 11%) and Pro-B cells/Pre-B cell fraction (0 to 9%). In accordance with French *et al*, we found no expression of IgM after the 21-day culture on MS-5 stroma neither from UCB CD34$^+$ cells nor from hPSC. Indeed, French *et al*, show on their results that another 21-day culture on MS-5 stroma with sorted CD45$^+$-derived cells is required to observe IgM expression [14]. IgM expression needs a sufficient B cell lineage engagement and a *RAG1* expression for the immunoglobulin gene rearrangement to occur. While we observed *RAG1* expression and Pre-BCR formation with genes coding for the two subunits of the surrogate light chain and the expression of the CD79a, component of the BCR, we obtained few B cells and they were blocked at a pre-B cells stage. This few amount of B cells did not permit a cell sorting to perform any additional and functional studies which highlights the need to improve HSC commitment in order to obtain proper lymphoid progenitors. Indeed a higher rate of HSC and hematopoietic progenitors will enhance B cell proportion and help optimizing B cell differentiation protocol onto MS-5 stroma with a sufficient amount to allow an efficient B cell generation at Day 21 and an IgM expression at Day 42 of B cell differentiation onto MS-5 stroma.

In their protocol, French *et al*. observed B cell differentiation using one hIPS cell line previously generated from fibroblasts. However, their group did not reach B cell generation at Day 21 from H1 hESC [17] while Vodyanick *et al*. demonstrated B cell generation with both H1 and H9 hESC [35] using the same protocol. In addition, Larbi *et al*. described B cell generation from H1 hESC with a three-step protocol where they first generated hematopoietic progenitors through embryoid body formation and then differentiated them into B cells by two successive

steps onto MS-5 stroma [29]. It must be taken into consideration that while they are largely similar, hIPSC may differ from their native hESC counterpart mostly in term of genetic and epigenetic expression [36–38]. Moreover, hIPSC may retain epigenetic memory of their original tissue cell type [39, 40] and their culture has been associated with the acquisition of mutations [41] and loss of imprinting [42], which can result in different differentiation potentials. These different differentiation abilities might be overcome in future study with the use of different cell line, especially hIPSC derided from B cells [43] which might differentiate into B cell with a better efficiency. To optimize hPSC differentiation toward an efficient lymphoid lineage commitment, high proportion of hematopoietic progenitors and their in-depth characterization, notably markers to differentiate primitive and definitive progenitors, are needed. To increase hematopoietic progenitors' proportion, a focus on the mesoderm progenitors generated during the first days is worth considering their cell fate differ. Indeed, we would prefer engage hPSC exclusively into posterior-primitive-streak hemogenic mesoderm nor mid-primitive-streak cardiogenic mesoderm which give rise to blood cell lineage vs cardiogenic cell fate respectively [44–46]. Specific markers of hematopoietic progenitors would also help following and comparing the differentiation efficiency according the protocol and the cell line. The expression of an intermediate level of both CD45 and CD31 might be a goal to reach, considering that UCB CD34$^+$ cells have this phenotype and that they efficiently differentiate into B cells onto MS-5 stroma (S3 Fig).

While considerable effort has been made in defining the hematopoietic progenitors and the signals needed for specific lineage engagement, we still need to better understand the hematopoietic process. Our work bring insights in *in vitro* B cell differentiation and in the characterization of hematopoietic progenitors. Such characterization is also high concern in hematopoietic stem cells transplantation. The design of robust protocol for lymphoid differentiation would help to better understand the hematopoietic process and would open avenues in future cell-based therapy.

## Supporting information

**S1 Fig. CD34$^+$ and B cell differentiation from H1 and H9 hESC are similar and cytokines at 10 or 25ng/ml during CD34$^+$ differentiation give the same differentiation results.**
(EPS)

**S2 Fig. A majority of CD45$^+$ cells might express CD31$^{int}$CD45$^{int}$ markers for B cells differentiation.**
(EPS)

**S3 Fig. Key steps during B cell differentiation from hPSC.**
(EPS)

## Acknowledgments

FD, LD, and SB designed the study. FD and AG carried out the experiments. FD, AG, FH, SC, LD and SB analyzed the data. LF, LT, JM-H and MC contributed for their help and advices during the study. FD, AG, FH, SC, LD and SB drafted and revised the paper. We thank iPSCDTC, TRIP and GenoCellEdit core facilities. All authors approved the final version of the manuscript.

## Author Contributions

**Conceptualization:** Fabienne Haspot, Sophie Conchon, Laurent David, Sophie Brouard.

**Data curation:** Florian Dubois, Léa Flippe, Jean-Marie Heslan, Laurent Tesson, Mélanie Chesneau.

**Formal analysis:** Florian Dubois, Anne Gaignerie.

**Funding acquisition:** Laurent David, Sophie Brouard.

**Investigation:** Florian Dubois, Laurent Tesson, Fabienne Haspot, Sophie Conchon, Laurent David, Sophie Brouard.

**Methodology:** Florian Dubois, Jean-Marie Heslan, Laurent Tesson, Fabienne Haspot, Sophie Conchon, Laurent David, Sophie Brouard.

**Project administration:** Sophie Conchon, Laurent David, Sophie Brouard.

**Resources:** Laurent David, Sophie Brouard.

**Software:** Florian Dubois.

**Supervision:** Florian Dubois, Fabienne Haspot, Sophie Conchon, Laurent David, Sophie Brouard.

**Validation:** Florian Dubois, Fabienne Haspot, Sophie Conchon, Laurent David, Sophie Brouard.

**Writing – original draft:** Florian Dubois, Fabienne Haspot, Sophie Conchon, Laurent David, Sophie Brouard.

**Writing – review & editing:** Florian Dubois, Anne Gaignerie, Fabienne Haspot, Sophie Conchon, Laurent David, Sophie Brouard.

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
