## [Decision Letter · Decision Letter 0]

27 Aug 2020

PONE-D-20-22523

Toward a better definition of hematopoietic progenitors suitable for B cell differentiation

PLOS ONE

Dear Dr. Dubois,

Thank you for submitting your manuscript to PLOS ONE. After careful consideration, we feel that it has merit but does not fully meet PLOS ONE’s publication criteria as it currently stands. Therefore, we invite you to submit a revised version of the manuscript that addresses the points raised during the review process.

The reviewers are concerned about the sample sizes and B cell characterization. Please address the comments accordingly during the revision. 

We look forward to receiving your revised manuscript.

Kind regards,

Xiaoping Bao, Ph.D.

Academic Editor

PLOS ONE

Journal Requirements:

4. Please include a copy of Table 1 which you refer to in your text on page 9

Reviewers' comments:

Reviewer's Responses to Questions

**Comments to the Author**

1. Is the manuscript technically sound, and do the data support the conclusions?

Reviewer #1: Yes

Reviewer #2: Partly

2. Has the statistical analysis been performed appropriately and rigorously? 

Reviewer #1: N/A

Reviewer #2: No

3. Have the authors made all data underlying the findings in their manuscript fully available?

Reviewer #1: Yes

Reviewer #2: Yes

4. Is the manuscript presented in an intelligible fashion and written in standard English?

Reviewer #1: Yes

Reviewer #2: Yes

5. Review Comments to the Author

Reviewer #1: The authors compared CD34+ progenitors derived from cord blood and human ESC. They identified CD31(int)CD45(int) as efficient hematopoietic progenitors suitable for B cell differentiation. Interestingly, human PSCs are less efficient to generate B cells than cord blood stem cells.

Major concerns:

1. In many experiments or conditions, there is only 1 independent experiment and there are no solid statistical comparison to reach any conclusion.

2. In the first 10 days derivation of CD34+ cell from hPSCs, although the authors tried to play with different concentrations or combinations of cytokines to get the highest efficiency, the transition from hPSCs into specific mesoderm progenitors and then differentiate into CD34+ progenitors are overlooked. Certain mesoderm progenitors (i.e., anterior or posterior) favor the hematopoietic differentiation.

3. The authors discussed a lot of efficienty difference and the functional characterization of the differentiated B cells except for B cell markers is missing

Reviewer #2: In this manuscript, Florian Dubois et al. reported that they were interested in the relationship between B cells and hPSCs-derived hematopoietic stem cells. They differentiated the hPSCs into hematopoietic stem cells through the OP9 stromal cell, and the B cells were generated on MS-5 stromal cells. They investigated the influence of various cytokine conditions on the hematopoietic stem cell generation, and they highlight CD31intCD45int phenotype as a possible marker of hematopoietic progenitors suitable for B cell differentiation.

This manuscript should be revised as below.

Major modification:

(1) At the introduction part, they authors should make more illustration about the relationship between B cells and hematopoietic stem cells, especially the published protocols for B cell differentiation.

(2) At the experiment section, the OP9 cells were cultured in αMEM medium instead of aMEM medium.

(3) The authors should perform further characterization on the iPSC-derived hematopoietic stem/progenitor cells, the characterization is not enough.

(4) As a biological experiment data, the authors should make additional statistical analysis, for example, Fig 2-7.

(5) In order to make this manuscript better, the authors should improve the layout quality of the Figures, especially Fig 5.

(6) The authors should make a scheme illustration to present their key ideas or results.

6. PLOS authors have the option to publish the peer review history of their article (what does this mean?). If published, this will include your full peer review and any attached files.

Reviewer #1: No

Reviewer #2: **Yes: **Yun Chang

---

## [Author Response · Author response to Decision Letter 0]

12 Nov 2020

Reviewer #1: The authors compared CD34+ progenitors derived from cord blood and human ESC. They identified CD31(int)CD45(int) as efficient hematopoietic progenitors suitable for B cell differentiation. Interestingly, human PSCs are less efficient to generate B cells than cord blood stem cells.

Major concerns: 

1. In many experiments or conditions, there is only 1 independent experiment and there are no solid statistical comparison to reach any conclusion.

For more clarity, we chose to only show the experiments performed with one cytokine concentration of 10ng/mL in Figure 2B and 2C. Nevertheless, this experiment with the condition D was reproduced several times with cytokines used at 25ng/mL (Figure 3A, n=3). As mentioned by reviewer 2, we performed statistical tests and confirmed statistical difference between the conditions and the CTRL in the figure 2, 3 and S1.

2. In the first 10 days derivation of CD34+ cell from hPSCs, although the authors tried to play with different concentrations or combinations of cytokines to get the highest efficiency, the transition from hPSCs into specific mesoderm progenitors and then differentiate into CD34+ progenitors are overlooked. Certain mesoderm progenitors (i.e., anterior or posterior) favor the hematopoietic differentiation.

Regarding OP9 stromal cells favor hematopoietic differentiation, we only focus about the hematopoietic progenitor generation. Nevertheless, we will look forward to check if there is a lack of mesoderm progenitors susceptible for hematopoietic differentiation in our model.

3. The authors discussed a lot of efficienty difference and the functional characterization of the differentiated B cells except for B cell markers is missing

During our differentiation, we show that hematopoietic progenitors follow the specific stage-differentiation of B cells both in term of molecular and transcriptional marker. However, the few amount of B cells generated did not permit any sorting and functional characterisation, we first need to optimise the protocol and reach a sufficient number of B cells. 

Reviewer #2: In this manuscript, Florian Dubois et al. reported that they were interested in the relationship between B cells and hPSCs-derived hematopoietic stem cells. They differentiated the hPSCs into hematopoietic stem cells through the OP9stromal cell, and the B cells were generated on MS-5 stromal cells. They investigated the influence of various cytokine conditions on the hematopoietic stem cell generation, and they highlight CD31intCD45int phenotype as a possible marker of hematopoietic progenitors suitable for B cell differentiation. This manuscript should be revised as below.

Major modification:

(1) At the introduction part, they authors should make more illustration about the relationship between B cells and hematopoietic stem cells, especially the published protocols for B cell differentiation.

Reviewer 2 is right, we highlighted this point in the introduction section (l.50).

(2) At the experiment section, the OP9 cells were cultured in αMEM medium instead of aMEM medium.

We double-checked and we confirmed that OP9 cells were cultured in the right �MEM medium (without nucleoside) supplemented with fetal bovine serum to a final concentration of 20% as recommended by ATCC; Vodyanick et al., 2005; Carpenter et al., 2011; and French et al., 2015. 

(3) The authors should perform further characterization on the iPSC-derived hematopoietic stem/progenitor cells, the characterization is not enough.

Because of the actual situation due to research shut down during the Covid19 pandemic, we were not able to performed new B cell differentiation.

(4) As a biological experiment data, the authors should make additional statistical analysis, for example, Fig 2-7.

Reviewer 2 is right, when possible (n > 3), statistical analysis were performed. We thus updated the material and method section (l.138) and the figures 2, 3 and S1.

(5) In order to make this manuscript better, the authors should improve the layout quality of the Figures, especially Fig 5.

We improved the layout quality of all figures.

(6) The authors should make a scheme illustration to present their key ideas or results.

We added a scheme illustration of the B cell differentiation key steps and its reference in the discussion section (l.370).

---

## [Decision Letter · Decision Letter 1]

30 Nov 2020

Toward a better definition of hematopoietic progenitors suitable for B cell differentiation

PONE-D-20-22523R1

Dear Dr. Dubois,

We’re pleased to inform you that your manuscript has been judged scientifically suitable for publication and will be formally accepted for publication once it meets all outstanding technical requirements.

Kind regards,

Xiaoping Bao, Ph.D.

Academic Editor

PLOS ONE

Additional Editor Comments (optional):

Reviewers' comments:

Reviewer's Responses to Questions

**Comments to the Author**

1. If the authors have adequately addressed your comments raised in a previous round of review and you feel that this manuscript is now acceptable for publication, you may indicate that here to bypass the “Comments to the Author” section, enter your conflict of interest statement in the “Confidential to Editor” section, and submit your "Accept" recommendation.

Reviewer #1: All comments have been addressed

Reviewer #2: All comments have been addressed

2. Is the manuscript technically sound, and do the data support the conclusions?

Reviewer #1: Yes

Reviewer #2: Yes

3. Has the statistical analysis been performed appropriately and rigorously? 

Reviewer #1: Yes

Reviewer #2: Yes

4. Have the authors made all data underlying the findings in their manuscript fully available?

Reviewer #1: Yes

Reviewer #2: Yes

5. Is the manuscript presented in an intelligible fashion and written in standard English?

Reviewer #1: Yes

Reviewer #2: Yes

6. Review Comments to the Author

Reviewer #1: the author addressed my questions and I recommend to publish this manuscript in the journal of plos one.

Reviewer #2: The authors have answered all of the questions.This manuscript was in good quality for publication.

7. PLOS authors have the option to publish the peer review history of their article (what does this mean?). If published, this will include your full peer review and any attached files.

Reviewer #1: No

Reviewer #2: **Yes: **YUN CHANG

---

## [Editor Report · Acceptance letter]

4 Dec 2020

PONE-D-20-22523R1 

Toward a better definition of hematopoietic progenitors suitable for B cell differentiation 

Dear Dr. Dubois:

I'm pleased to inform you that your manuscript has been deemed suitable for publication in PLOS ONE. Congratulations! Your manuscript is now with our production department. 

Kind regards, 

on behalf of

Dr. Xiaoping Bao 

Academic Editor

PLOS ONE